# Value-based person-centred integrated care for frail elderly living at home: a quasi-experimental evaluation using multicriteria decision analysis

Maaike Hoedemakers ,[1] Milad Karimi,[1,2] Fenna Leijten,[1,3] Lucas Goossens,[1] Kamrul Islam,[4,5] Apostolos Tsiachristas ,[6] Maureen Rutten-van Molken [1]

¹Erasmus School of Health Policy & Management, Erasmus Universiteit Rotterdam, Rotterdam, The Netherlands
²OPEN Health, Rotterdam, The Netherlands
³Staff Defence Healthcare Organisation, Ministry of Defence, Utrecht, The Netherlands
⁴Department of Economics, University of Bergen, Bergen, Norway
⁵Social Sciences, NORCE Norwegian Research Centre AS, Bergen, Norway
⁶Nuffield Department of Population Health, University of Oxford, Oxford, UK

**Correspondence to**
Dr Maureen Rutten-van Molken;
m.rutten@eshpm.eur.nl

## ABSTRACT

**Objective** To evaluate the value of the person-centred, integrated care programme Care Chain Frail Elderly (CCFE) compared with usual care, using multicriteria decision analysis (MCDA).

**Design** In a 12-month quasi-experimental study, triple-aim outcomes were measured at 0, 6 and 12 months by trained interviewers during home-visits.

**Setting** Primary care, community-based elderly care.

**Participants** 384 community-dwelling frail elderly were enrolled. The 12-month completion rate was 70% in both groups. Propensity score matching was used to balance age, gender, marital status, living situation, education, smoking status and 3 month costs prior to baseline between the two groups.

**Intervention** The CCFE is an integrated care programme with unique features like the presence of the elderly and informal caregiver at the multidisciplinary team meetings, and a bundled payment.

**Primary and secondary outcomes measures** The MCDA results in weighted overall value scores that combines the performance on physical functioning, psychological well-being, social relationships and participation, enjoyment of life, resilience, person-centredness, continuity of care and costs, with importance weights of patients, informal caregivers, professionals, payers and policy-makers.

**Results** At 6 months, the overall value scores of CCFE were higher in all stakeholder groups, driven by enjoyment of life (standardised performance scores 0.729 vs 0.685) and person-centredness (0.749 vs 0.663). At 12 months, the overall value scores in both groups were similar from a patient's perspective, slightly higher for CCFE from an informal caregiver's and professional's perspective, and lower for CCFE from a payer's and policy-maker's perspective. The latter was driven by a worse performance on physical functioning (0.682 vs 0.731) and higher costs (€22 816 vs €20 680).

**Conclusions** The MCDA indicated that the CCFE is the preferred way of delivering care to frail elderly at 6 months. However, at 12 months, MCDA results showed little difference from the perspective of patients, informal caregivers and professionals, while payers and policy-makers seemed to prefer usual care.

## STRENGTHS AND LIMITATIONS OF THIS STUDY

⇒ We implemented a novel multicriteria decision analysis framework to evaluate a person-centred integrated care programme for frail elderly.
⇒ We measured a broad range of triple-aim outcomes at three time points, among a large and difficult to reach group of the frailest elderly living at home.
⇒ The choice of outcome measures was largely driven by focus groups involving frail elderly.
⇒ The different outcome measures were weighted by their importance, from multiple perspectives, to calculate overall value scores.
⇒ Due to the choice of measuring patient-reported outcomes, participation was too burdensome or impossible for frail elderly with cognitive impairments.

## INTRODUCTION

The Care Chain Frail Elderly (CCFE) is an integrated care programme for community-dwelling frail elderly in the Netherlands that offers person-centred care, coordination, and case management to support elderly in living at home for as long as possible.[1] It was developed as part of a movement towards decentralisation of long-term care, increased self-sufficiency, and societal participation that was stimulated by the Dutch government to maintain affordability of elderly care. This movement led to a reform of the long-term care sector in 2015.[2] The number of nursing and residential homes was reduced considerably and access to nursing homes was restricted to those in need of 24-hour care. Municipalities became responsible for the provision of homecare and social support, and health insurers for financing nursing care at home. While the importance of homecare was growing, this sector was confronted with significant budget cuts.[3] The reform accelerated the development of integrated-care initiatives, which were spearheaded by

the National Care for the Elderly-Programme commissioned by the Ministry of Health.[4 5]

Integrated care for older people often comprises similar components, namely, comprehensive assessment, individualised care planning, case management and multidisciplinary team meetings.[6 7] All of these elements are also present in the CCFE. In addition, the intervention is unique in several respects. First, the older person and their informal caregiver are present at the multidisciplinary team meetings where the results of the comprehensive assessment and the individual care plan are discussed. Second, the CCFE is financed by a bundled payment aiming to stimulate collaboration between professionals. The bundled payment is a fixed amount of money per patient that covers all services provided by the general practitioner (GP), nurse–practitioner and physician assistant, regardless of diagnoses, medication review by the pharmacist, telephone consultation by the geriatrician, non-individual-patient-related activities such as building a community network, and overhead. Third, it targets the top 1% frailest elderly registered in a GP practice who are living at home with complex care needs, using a case-finding approach. The CCFE aims to better integrate care across sectors and build a network of support around the patient, and thereby to improve their physical, mental, social health and well-being and experience with care. Ideally, this also reduces secondary care and residential long-term care utilisation and thereby costs. The CCFE has previously been described in detail elsewhere.[1]

Although such integrated-care programmes are designed to meet the older person's needs, previous studies show mixed (cost-)effectiveness results.[8–10] A potential explanation is that common methods to assess effectiveness tend to focus on measuring traditional outcomes, such as physical functioning, whereas that is not the primary focus of these programmes.[8 11] Cost-effectiveness analyses measure quality-adjusted life-year (QALY) gains, which is also less appropriate because integrated care for frail elderly focuses more on well-being than on survival and health-related quality of life.[12] For example, elderly are stimulated to visit outpatient day-care activities to enhance their social participation, or their experience with care is improved by individual care planning and helping them navigate through the health-care and socialcare system. Accordingly, empirical evaluations should include these domains to accurately value the potential benefits of an intervention.[13 14]

Therefore, we adopted a broader evaluation method, namely multicriteria decision analysis (MCDA) to evaluate the CCFE. MCDA is innovative in elderly care because it synthesises a wide variety of outcome measures, in this case patient-reported health and well-being measures (patient-reported outcomes, PROMs), experience with care measures (PREMs) and costs. Together, they cover the triple aim.[15] In the MCDA outcomes are weighted by their importance according to five stakeholder groups: patients, informal caregivers, professionals, policy-makers and payers. The weighted performance scores are aggregated into overall value scores.[16]

## METHODS

### Study population

The study population consisted of community-dwelling frail elderly, where frailty is defined as being in need of complex care due to loss of functional abilities and control over one's life. To participate in the intervention, they had to be registered at a GP practice from one of three care groups (ie, an association of primary care providers that cooperate in the provision of chronic care) that offer the CCFE. They had to be able to comprehend study information and answer questions, either independently or with the help of an informal caregiver or trained interviewer.

### Intervention

A primary care team consisting of the GP, nurse–practitioner, and district nurse, identifies potential candidates for the CCFE using a case-finding approach. To be included in the CCFE, the health insurer does not require a specific diagnosis or the use of a screening tool. They trust that the GP and nurse practitioner know their patients and are most suited to select the elderly that may benefit from the care programme. Furthermore, there is an agreement with the health insurer that only the top 1% frail elderly of a GP practice will be included in the programme. Subsequently, the nurse–practitioner visits the older person at home and performs a comprehensive geriatric assessment of the needs, capabilities and preferences in the physical, psychological, cognitive and social domains. Depending on this assessment, the nurse–practitioner consults professionals in the community (eg, physical therapists, occupational therapists, social workers, elderly care physicians, geriatricians, dementia case workers) and arranges informal care support. Together with the frail elderly and informal caregiver, the nurse–practitioner drafts an individual care plan, largely driven by the elderly's personal goals.

The nurse–practitioner organises multidisciplinary team meetings to discuss the individual care plan with all professionals involved. The frail elderly and their informal caregiver participate in these meetings. During the first meeting a case manager is assigned, a role mostly taken up by the nurse–practitioner. The case manager is the main contact point, monitors the execution of the individual care plan, and further adapts the care to the patient's wishes and additional needs. Once enrolled, an elderly person usually stays in the CCFE until nursing home admission or if they pass away.

To support collaboration between professionals, the CCFE uses 'Care2U', a secured ICT-platform to share information. Professionals have different degrees of access and the elderly must approve access. Care2U includes the individual care plan and is used by the nurse–practitioner

to monitor appointments and services of providers in the care chain.

The CCFE is financed by a bundled payment contract between each care group and the dominant health insurer in the region and is renegotiated every year. These care groups are legal entities of primary care providers who develop chronic care programmes, support the provision of these programmes and are contracted by the health insurer to coordinate chronic care in a region.

## Study design

The MCDA was conducted as part of a 12-month prospective quasi-experimental study comparing two parallel groups:

Intervention group included frail elderly enrolled into the CCFE between April 2017 and August 2018 and were recruited to participate in the study by either their GP or nurse–practitioner.

Control group consisted of frail elderly receiving usual care, recruited at GP practices from one of three participating care groups. These practices had not (yet) implemented the CCFE. To ensure a similar level of frailty in both groups, the GP practices in the control group applied the same case-finding approach. They were assisted by a GP specialised in elderly care that had experience with the CCFE.

Self-reported outcome and experience measures and care utilisation data were gathered at baseline and after 6 months and 12 months, during face-to-face interviews at the elderly's home, performed by trained interviewers. The interviewers were not involved in delivering any aspects of the intervention but were aware of which group the respondent belonged to. Data collection took place between April 2017 and August 2019.

## Outcome measures

Table 1 gives an overview of all outcome measures. These outcomes were selected based on a literature review, workshops with representatives from the five stakeholder groups, and focus groups with individuals with multimorbidity, and measured with validated questionnaires.[17]

## Costs

Healthcareand socialcare utilisation data were obtained with the institute for Medical Technology Assessment (iMTA) Medical Consumption Questionnaire, which includes questions about contacts with GP, nurse–practitioner, GP–assistants, physiotherapists and other

**Table 1** Outcome measures and instruments

| Core set* | Instrument to measure outcome | Scale |
|---|---|---|
| Health and well-being | | |
| Physical functioning | Activities of Daily Living (Katz-15)[37] | 0–15 (worst) |
| Psychological well-being | Mental Health Inventory[38] | 0–100 (best) |
| Enjoyment of life | Investigating Choice Experiments for the Preferences of Older People[39] | 1–4 (best) |
| Social relationships and participation | Impact on Participation & Autonomy, social life and relationships domain[40] | 0–28 (worst) |
| Resilience | Brief Resilience Scale[41] | 6–30 (best) |
| Experience of care | | |
| Person-centredness | Person-centred Coordinated Care Experience Questionnaire, experience of person-centred care domain[42] | 0–18 (best) |
| Continuity of care | Nijmegen Continuity Questionnaire, team and cross boundary continuity domain[43] + Client Perceptions of Coordination Questionnaire[44] | 1–5 (best) |
| Costs | | |
| Health, social, and informal care costs | iMTA Medical Consumption Questionnaire[18] | |
| Medication costs | Prescriptions in patient records extracted from GP information systems | |
| Bundled payments and chronic care programmes | Care chain information system 'Care2U' | |
| Additional outcomes | | |
| Autonomy | Pearlin Mastery Scale[45] | 7–35 (best) |
| Burden of medication | Living with Medicines Questionnaire[46] | 0–10 (worst) |

*The core set of outcomes was measured across all studies included in the SELFIE project. For these outcomes, weights were elicited, and these outcomes were included in the MCDA. The additional outcomes were not included in the MCDA.
GP, general practitioner; MCDA, multicriteria decision analysis; SELFIE, Sustainable intEgrated care modeLs for multi-morbidity: delivery, FInancing and performancE.

paramedical therapists, dieticians, psychologists, social workers, welfare workers and medical specialists, hospital admissions, rehabilitation, homecare, residential care and nursing homes, and informal care during the past 3 months.[18] Unit costs were largely based on reference prices from the Dutch Costing Manual.[19] Medication costs were based on prescription data from GP-information systems, which were combined with unit costs from Dutch drug database 'G-Standaard', using Anatomical Therapeutic Chemical (ATC) codes.[20] Programme costs of the CCFE were based on the bundled payment contracts between each care group and the dominant health insurer.[1] Elderly from the control group could participate in single-disease care programmes for diabetes, cardiovascular risk management or chronic obstructive pulmonary disease. The proportion of elderly participating in these programmes was obtained from Care2U and average prices of the bundled payments were obtained from the Dutch health claims database (Vektis).[21]

### Statistical analysis
#### Comparability of groups
We applied inverse probability weighting (IPW) to increase the comparability of the intervention and control groups at baseline.[22] The logistic regression model to estimate the propensity score included age, gender, marital status, living situation, educational level, smoking status and costs 3 months prior to baseline as a proxy for complexity. The propensity score $p$ reflects the estimated probability of an individual to be in the intervention group. By setting the weight for individuals in the intervention group to 1, and for individuals in the control group to $p/(1-p)$, we estimated the average treatment effect in the treated in the weighted mixed effect models described below.[23] To assess the comparability of the two groups, we calculated the mean percentage standardised bias, the Rubin's B (absolute standardised difference of the means of the linear index of the propensity score in the intervention group and matched controls), and the Rubin's R (ratio of intervention group and matched control group variances of the propensity score index). For sufficient balance, B should be less than 25% and R be between 0.5 and 2.[24]

#### Treatment effects
Treatment effects were estimated using weighted mixed effect models with a random intercept at individual level (as we had longitudinal data) and the following covariates: time, intervention, an interaction term for time and intervention, age, gender, marital status, living situation, educational level and smoking status. This combination of matching and regression adjustment has been shown to best reduce covariate imbalance between groups.[25] The mixed effects models were used to predict the mean scores of the outcome measures in both groups at each time point, assuming the control group had the same baseline score as the intervention group to directly compare both groups. All statistical analyses were performed using STATA V.16.1.

#### MCDA
In the MCDA, the predicted mean scores of the outcomes at 6 months and 12 months follow-up were standardised on a 0–1 scale to remove differences in measurement scales, using relative standardisation, see online supplemental appendix 1. For all outcomes in the MCDA, a higher score indicates better performance. The standardised outcomes were weighted by their importance and subsequently summed to obtain an overall value score for the intervention and control group separately. The relative importance-weights were elicited in an online weight elicitation study among patients, informal caregivers, professionals, payers and policy-makers, using a discrete choice experiment (DCE).[26] The relative weights of the outcome measures included in the MCDA by stakeholder group can be found in online supplemental appendix 2.

#### Sensitivity analysis
To assess the joint uncertainty in outcome scores and importance-weights, we performed a probabilistic sensitivity analysis using Monte Carlo simulation. We used Cholesky decomposition to obtain 10 000 replications of both the standardised outcomes and the importance weights. From this, we obtained the 95% CIs around the overall value scores for each stakeholder group. Additionally, we calculated the proportion of MCDA iterations in which the CCFE has a higher overall value score than usual care.

### Patient and public involvement
Patients were involved in the selection of outcome measures, which was largely based on focus groups as described above.[17] In multiple National Stakeholder Workshops held during the entire process of the study, stakeholders from the five stakeholder groups were asked to reflect on the outcome measures, the study design and data collection, and the results of the study. The study design was set up in close collaboration with care providers to ensure feasibility of the data collection. Before the start of data collection, the questionnaire was piloted in a frail older person. The authors disseminated results via conference presentations. Results of this study were also disseminated to participating care providers, patients and informal caregivers using communication methods other than scientific papers, that is, by email and newsletters disseminated at the GP practices.[27]

## RESULTS
### Respondents
Figure 1 presents the flow chart of patients included in the study. The retention rate at 12-month follow-up was 70% in both groups. Main reasons of lost to follow-up were also similar in both groups and included death, burden of study participation and cognitive incapacity.

Table 2 presents the baseline characteristics of respondents before and after IPW. After IPW, the matching statistics were within the desired range (Rubin's B<25%,

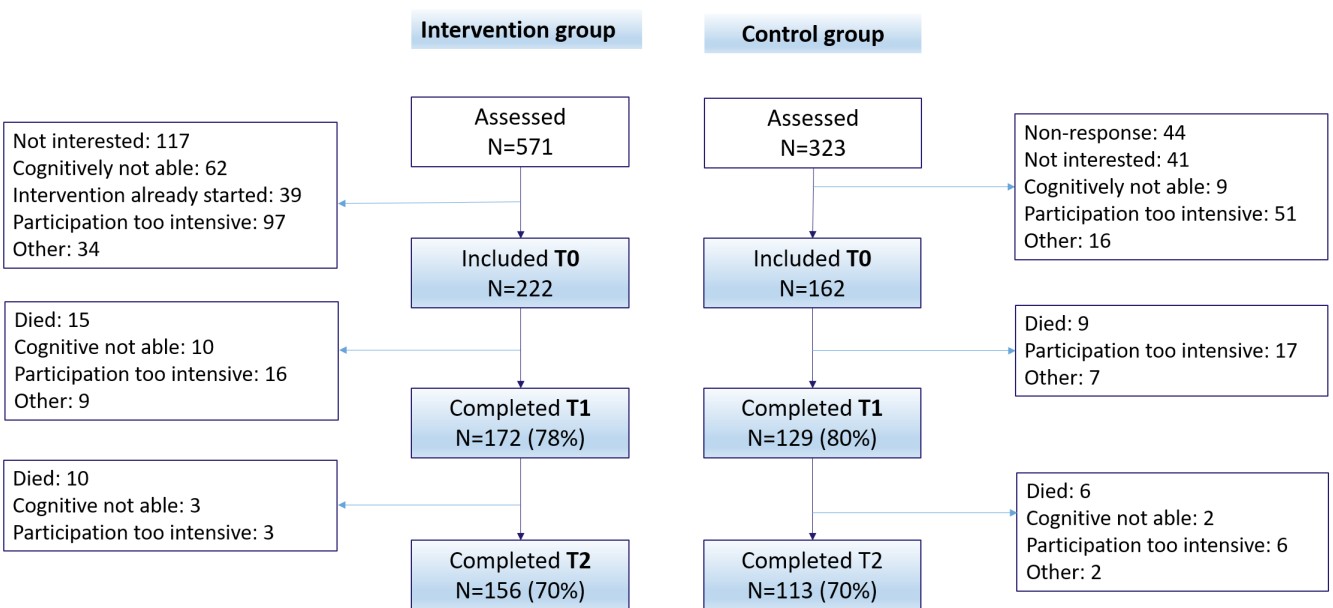

**Figure 1** Flow chart of patient participation.

Rubin's R 0.5–2). No substantial differences between the groups remained. The common support graph shows a good overlap in propensity scores (see online supplemental appendix 3).

## Treatment effects

Table 3 reports the estimated treatment effects of the CCFE at 6 and 12 months of follow-up. Results show that the CCFE improved person-centredness at both time points. At 6 months follow-up, physical functioning declined in both groups but even further in the intervention group. At 12 months follow-up, the CCFE performed worse on autonomy and burden of medication. The predicted mean performance scores of all outcomes on their natural scale can be found in online supplemental appendix 4.

## Costs

Table 4 presents details on the mean costs after inversed probability weighting at 6 and 12 months, from a healthcare and a societal perspective. From a healthcare perspective, after 6 months, costs were €751 higher in the intervention group than in the control group. After 12 months, the costs were €1796 higher. These differences were largely due to programme costs and costs of homecare.

When adopting a societal perspective, 6 months costs were €662 higher in the intervention group than in the control group, whereas 12 months costs were €2136 higher. Costs of informal care were slightly lower in the intervention group than in the control group after 6 months, but higher after 12 months.

## Multicriteria decision analysis

Table 5A,B presents the standardised outcome scores for the CCFE and usual care as well as these scores weighted according to each stakeholder's importance weights, at 6 and 12 months, respectively. The overall value scores show that all stakeholder groups preferred the CCFE over usual care at 6 months follow-up. This was driven by the performance scores of person-centredness and enjoyment of life, and the high importance-weight of the latter. In more than 75% of iterations the CCFE had a higher overall value score than usual care. At 12 months follow-up, the difference disappeared, and the probability that the CCFE had a higher overall value score dropped below 50% for payers and policy-makers. This was driven by worse scores in physical functioning and costs.

## DISCUSSION
### Main findings

The MCDA has shown that all stakeholders preferred the CCFE over usual care at 6 months with a likelihood of having a higher overall value score of over 75%. This was mainly driven by higher performance on enjoyment of life and person-centredness, and the great importance of the former outcome. Results became more diffuse at 12 months. Patients were indifferent, informal caregivers and professionals slightly favoured the CCFE, whereas payers and policy-makers demonstrated a slight preference for usual care. This was mainly due to a worse performance of the CCFE on physical functioning and costs. When looking at the disaggregated scores, person-centredness was consistently higher in the CCFE. Physical functioning deteriorated in the intervention group at 6 months, but this effect disappeared at 12 months. When some outcomes improve whereas others deteriorate, the current MCDA approach is a suitable method to aggregate them into overall value scores that vary depending on the importance that stakeholders assign to

**Table 2** Baseline characteristics before and after inversed probability weighting

| Baseline characteristics | CCFE N=222 | UC N=162 | SD | UC N=162 | SD |
|---|---|---|---|---|---|
| | Before IPW | | | After IPW | |
| Age, mean | 83.4 | 84.8 | −0.224 | 83.6 | −0.028 |
| Female (%) | 63.5 | 64.2 | −0.014 | 64.8 | −0.027 |
| Marital status (%) | | | | | |
| Single, never married | 3.6 | 3.7 | −0.005 | 2.6 | 0.054 |
| Married or living together | 44.6 | 40.7 | 0.078 | 45.0 | −0.008 |
| Widow(er) | 44.6 | 50.6 | −0.120 | 47.1 | −0.049 |
| Divorced | 7.2 | 4.9 | 0.095 | 5.4 | 0.078 |
| Living situation (%) | | | | | |
| Independent, alone | 54.1 | 61.1 | −0.143 | 55.2 | −0.024 |
| With others | 46.0 | 38.9 | 0.143 | 44.8 | 0.024 |
| Educational level (%) | | | | | |
| Low | 70.3 | 70.4 | −0.002 | 73.3 | −0.067 |
| Medium | 20.3 | 14.8 | 0.143 | 14.3 | 0.158 |
| High | 9.5 | 14.8 | −0.164 | 12.4 | −0.090 |
| Current smoker (%) | 14.4 | 8.6 | 0.181 | 13.8 | 0.019 |
| **Outcome measures at baseline, mean** | **Before IPW** | | | **After IPW** | |
| Physical functioning (0–15)* | 4.4 | 4.5 | −0.032 | 4.5 | −0.052 |
| Psychological well-being (0–100) | 71.4 | 71.7 | −0.018 | 70.6 | 0.040 |
| Enjoyment of life (1–4) | 2.8 | 2.9 | −0.137 | 2.9 | −0.143 |
| Social relationships and participation (0–28)* | 9.2 | 8.0 | 0.345 | 8.2 | 0.280 |
| Resilience (6–30) | 19.3 | 19.0 | 0.062 | 19.0 | 0.082 |
| Autonomy (7–35) | 22.3 | 22.1 | 0.054 | 21.9 | 0.098 |
| Burden of medication (0–10)* | 2.1 | 2.4 | −0.111 | 2.5 | −0.166 |
| Person-centredness (0–18) | 11.7 | 12.5 | −0.209 | 12.7 | −0.246 |
| Continuity of care (1–5) | 3.7 | 3.8 | −0.140 | 3.8 | −0.182 |
| Total costs 3 months prior to the study (€)* | 5453 | 5267 | 0.028 | 5631 | −0.026 |
| **Statistics to assess matching†** | **Before IPW** | **After IPW** | | | |
| Mean bias | 11.9 | 5.1 | | | |
| Rubin's B | 39.7 | 21.4 | | | |
| Rubin's R | 1.06 | 1.10 | | | |

SD= absolute standardised mean difference, also called absolute standardised bias.
*Higher score indicating worse performance.
†On variables used in Propensity Score Matching, that is, age, gender, marital status, living situation, educational level, smoking, total costs 3 months prior to the study.
CCFE, Care Chain Frail Elderly; IPW, inverse probability weighting; UC, usual care.

the different outcome measures. These results show that the CCFE is the preferred way of delivering care to frail elderly if improvements in enjoyment of life and person-centredness are considered more important than physical functioning and costs.

### Context and comparison with previous evaluations

As this is the first MCDA of a frail elderly programme, it is impossible to directly compare the value scores to other studies. MCDA also provides insight in the disaggregated effects of the CCFE, but these are hard to compare to other studies as well, due to the very frail target group of the CCFE, the different contexts in which the interventions are implemented, the different intervention components and outcome measures.[28 29] Regarding the context of the CCFE, we should stress that the programme was implemented in a setting with a strong primary care sector. GPs already have a history of collaborating with other primary care providers, for example, by working jointly

**Table 3** Treatment effects at 6-month and 12-month follow-up

| Core set of outcomes included in the MCDA | 6 months | | | 12 months | | |
|---|---|---|---|---|---|---|
| | Estimated change CCFE | Estimated Change UC | Diff in change Mean (95% CI)† | Estimated change CCFE | Estimated change UC | Diff in change Mean (95% CI)† |
| Health and well-being | | | | | | |
| Physical functioning (0–15)‡ | 0.74 | 0.27 | 0.47 (0.06 to 0.88)* | 1.33 | 0.95 | 0.39 (−0.10 to 0.87) |
| Psychological well-being (0–100) | −0.64 | −0.34 | −0.30 (−3.86 to 3.25) | −1.35 | −0.10 | −1.26 (−5.20 to 2.68) |
| Enjoyment of life (1–4) | 0.10 | −0.08 | 0.18 (−0.02 to 0.38) | 0.05 | 0.02 | 0.04 (−0.18 to 0.25) |
| Social relationships and participation (0–28)‡ | 0.11 | 0.41 | −0.31 (−1.04 to 0.43) | 0.26 | 0.61 | −0.36 (−1.25 to 0.54) |
| Resilience (6–30) | −0.02 | −0.21 | 0.19 (−0.63 to 1.01) | 0.03 | 0.11 | −0.08 (−1.01 to 0.85) |
| Experience of care | | | | | | |
| Person-centredness (0–18) | 1.10 | −0.38 | 1.48 (0.57 to 2.38)* | 1.33 | 0.00 | 1.33 (0.18 to 2.49)* |
| Continuity of care (1–5) | 0.10 | −0.02 | 0.12 (−0.05 to 0.28) | 0.16 | 0.03 | 0.13 (−0.05 to 0.31) |
| Costs | | | | | | |
| Total health and social care costs‡ | 5405 | 4745 | 660 (−1650 to 2970) | 17 223 | 15 206 | 2017 (−2361 to 6395) |
| **Additional frail elderly-specific outcomes** | **Estimated change CCFE** | **Estimated change UC** | **Diff in change Mean (95% CI)†** | **Estimated change CCFE** | **Estimated change UC** | **Diff in change Mean (95% CI)†** |
| Autonomy (7–35) | −0.50 | 0.02 | −0.52 (−1.38 to 0.34) | −0.49 | 0.55 | −1.04 (−1.95 to −0.14)* |
| Burden of medication (0–10)‡ | 0.13 | −0.21 | 0.34 (−0.36 to 1.04) | 0.18 | −0.61 | 0.78 (0.04 to 1.53)* |

*P<0.05.
†Based on robust SE.
‡Higher score indicating worse performance.
CCFE, Care Chain Frail Elderly; MCDA, multicriteria decision analysis; UC, Usual care.

in health centres. Hence, setting up community networks and collaborating in wider multidisciplinary teams was not such a big step. When implementing similar interventions in countries with a less strong primary care system, collaboration may require more efforts. On the other hand, the potential for savings due to the intervention might be higher in countries where the GPs does not act as a gatekeeper to secondary care, as a programme like the CCFE could substitute more secondary care services by primary care services.

Despite differences between studies, most previous studies did not find effects on physical functioning.[8 30] This may be expected as these programmes rarely aim to achieve improvements in this domain. As the CCFE aimed to improve experience with care, the sustained improvement in person-centredness found in this study suggests that the programme has fulfilled that aim.[1] The worsening in autonomy and burden of medication seems counterintuitive. A possible explanation for the deterioration in autonomy is that as elderly in the CCFE were confronted with their frailty, for example, by discussing their needs for support, they became more aware of their loss of control. This could have led to lower autonomy scores in the intervention group, especially due to the self-report method; measurement by a third party might have led to different scores.[31] The higher medication burden in the intervention group could be explained by alterations in medication after the medication review, which may have led to (temporary) side effects.

One could question the relevance of finding a sustained effect on an outcome (ie, person-centredness) that was less highly valued than other outcomes in the DCE. This may raise the question whether the aim of the CCFE was well-targeted. However, weights were derived from a DCE that asks respondents to choose between two hypothetical care programmes, which gives them the opportunity to trade-off person-centredness for, for example, improved physical functioning. Even though this is likely to be an appropriate reflection of their preferences if all options were open, in real life this trade-off may no longer exist, because improving physical functioning might not be possible anymore. Hence, there is a discrepancy between what is important to a patient and what is feasible in practice.

### Strengths and limitations

One of the strengths of our study was its controlled study design. Defining an appropriate control group to evaluate ongoing programmes for frail elderly is a challenge.[8 32] In this study, a potential limitation concerns the case finding approach to identify the target population of both groups, namely frail elderly in need of complex care due to loss of functional abilities and control over one's life. In the intervention group this was done by GPs offering the programme and in the control group by GPs not offering the programme. To ensure a similar level of frailty in both groups, the latter GPs were assisted by a GP specialised in elderly care. This has been successful as the baseline characteristics of both groups were quite similar. It is especially important in evaluating frail elderly care, as there is commonly little room for actual improvements in

**Table 4** Mean costs (€) after 6 months and 12 months of follow-up (after IPW)

| Cost category | 6 months | | | 12 months | | |
|---|---|---|---|---|---|---|
| | Mean costs (SD) CCFE (n=172) | Mean costs (SD) UC (n=129) | Difference between means (SE) | Mean costs (SD) CCFE (n=156) | Mean costs (SD) UC (n=113) | Difference between means (SE) |
| Chronic care programme(s)* | 534 (-) | 72 (-) | 462 (-) | 1068 (-) | 143 (-) | 925 (-) |
| Homecare | 3289 (4371) | 3158 (4016) | 131 (481) | 7597 (8474) | 6330 (7680) | 1267 (980) |
| Long-term care admissions | 969 (5411) | 766 (4980) | 203 (589) | 1624 (7959) | 2051 (10025) | −427 (1133) |
| Hospital admissions | 670 (2670) | 485 (1833) | 185 (259) | 1199 (3603) | 1119 (2992) | 80 (403) |
| Emergency room visits | 112 (308) | 116 (341) | −4 (38) | 179 (372) | 196 (418) | −17 (49) |
| Outpatient day-care activities | 235 (1180) | 310 (1481) | −74 (170) | 627 (2642) | 408 (1846) | 218 (285) |
| Medical specialist care | 338 (619) | 317 (408) | 21 (60) | 631 (789) | 609 (615) | 23 (86) |
| Paramedical care (eg, physiotherapist) | 529 (636) | 481 (548) | 48 (72) | 1059 (1113) | 1005 (1133) | 55 (145) |
| GP† | 21 (27) | 288 (272) | −267 (28) | 25 (34) | 492 (394) | −467 (44) |
| Medication | 363 (662) | 318 (551) | 45 (65) | 744 (1232) | 605 (927) | 139 (124) |
| Subtotal costs from healthcare perspective | 7060 (8441) | 6310 (7671) | 751 (922) | 14753 (14498) | 12957 (14185) | 1796 (1777) |
| Informal care | 3690 (5780) | 3779 (7318) | −89 (891) | 8063 (12436) | 7723 (13643) | 340 (1783) |
| Total costs from societal perspective | 10750 (10381) | 10089 (11003) | 662 (1309) | 22816 (19050) | 20680 (20251) | 2136 (2551) |

*Costs for chronic care program(s) is an average estimation which is the same for each respondent in their respective group.
†Costs for GP-care in the intervention group are largely included in the costs of the chronic care programme.
CCFE, Care Chain Frail Elderly; GP, general practitioner; IPW, inverse probability weighting; UC, usual care.

health, and prevention or delay of deterioration can only be shown in comparison with a control group.

Another strength was the data collection on a broad range of PROMs and PREMs by interviewers who made a total of 954 home visits, which was a major endeavour. Collecting patient-reported data did limit the generalisability of the results, as some frail elderly could not participate in the evaluation because that was too burdensome or impossible. Hence, the frailest among the elderly were not represented in this study, especially not those with dementia. The attrition rate in our study was relatively low, ie, 30% across both groups at 12 months. We did observe that respondents in the control group that were lost to follow-up were slightly older and had worse physical functioning at baseline compared with non-drop-outs in the control group (and overall) (see online supplemental appendix 5). This may have led to an underestimation of the treatment effect.

A further strength of our study was the detailed cost analysis, even including costs of medication, social care and informal care, which are often excluded from other studies.[8] This analysis showed a cost increase reflecting the greater investment of resources to support frail elderly in ageing in place which is of great importance to many older persons.

In the design and reporting on the MCDA we followed the good practice guidelines as laid out by the ISPOR MCDA-taskforce.[33] Strengths of MCDA are that it enables explicit, transparent and accountable decision-making, that is, for every decision what was valued most and by whom can be tracked down as well as whether this was due to improvements in certain domains or a higher relative importance of a particular outcome. Furthermore, MCDA makes it possible to include additional elements of value that go beyond health or QALYs, which is especially important for complex interventions with multiple aims such as improving well-being and experience with care. However, the consequence is that we may favour interventions that achieve improvements in these outcomes above interventions that have greater health outcomes. This may be justifiable for elderly care. Such an argument would raise another point that is debated in MCDA,

**Table 5** (A) Value scores in the multicriteria decision analysis at 6 months, using DCE weights. (B) Value scores in the multicriteria decision analysis at 12 months, using DCE weights

**(A)**

| Outcome measures | Standardised Performance score* | | Patients Weighted score | | Partners Weighted score | | Professionals Weighted score | | Payers Weighted score | | Policy-makers Weighted score | |
|---|---|---|---|---|---|---|---|---|---|---|---|---|
| | CCFE | UC | CCFE | UC | CCFE | UC | CCFE | UC | CCFE | UC | CCFE | UC |
| Physical functioning | 0.672 | 0.74 | 0.108 | 0.118 | 0.074 | 0.081 | 0.081 | 0.089 | 0.094 | 0.104 | 0.094 | 0.104 |
| Psychological well-being | 0.706 | 0.709 | 0.12 | 0.12 | 0.106 | 0.106 | 0.127 | 0.128 | 0.127 | 0.128 | 0.106 | 0.106 |
| Enjoyment of life | 0.729 | 0.685 | 0.168 | 0.157 | 0.182 | 0.171 | 0.16 | 0.151 | 0.175 | 0.164 | 0.16 | 0.151 |
| Social relationships and participation | 0.718 | 0.696 | 0.057 | 0.056 | 0.065 | 0.063 | 0.079 | 0.077 | 0.072 | 0.07 | 0.072 | 0.07 |
| Resilience | 0.711 | 0.704 | 0.107 | 0.106 | 0.099 | 0.099 | 0.092 | 0.091 | 0.078 | 0.077 | 0.099 | 0.099 |
| Person-centredness | 0.749 | 0.663 | 0.06 | 0.053 | 0.06 | 0.053 | 0.06 | 0.053 | 0.045 | 0.04 | 0.06 | 0.053 |
| Continuity of care | 0.718 | 0.696 | 0.072 | 0.07 | 0.086 | 0.083 | 0.079 | 0.077 | 0.057 | 0.056 | 0.072 | 0.07 |
| Costs (societal care perspective) | 0.685 | 0.729 | 0.021 | 0.022 | 0.041 | 0.044 | 0.041 | 0.044 | 0.055 | 0.058 | 0.048 | 0.051 |
| Overall value scores | | | 0.711 | 0.702 | 0.713 | 0.7 | 0.719 | 0.708 | 0.703 | 0.696 | 0.711 | 0.702 |
| 95% CI* | | | 0.702 to 0.721 | 0.692 to 0.712 | 0.703 to 0.723 | 0.690 to 0.710 | 0.702 to 0.722 | 0.691 to 0.711 | 0.699 to 0.721 | 0.692 to 0.714 | 0.700 to 0.721 | 0.692 to 0.712 |
| % CCFE >UC* | | | 82 | | 90 | | 87 | | 75 | | 81 | |

**Table 5** Continued

(B)

| Outcome measures | Standardised Performance score* | | Patients Weighted score | | Partners Weighted score | | Professionals Weighted score | | Payers Weighted score | | Policy-makers Weighted score | |
|---|---|---|---|---|---|---|---|---|---|---|---|---|
| | CCFE | UC | CCFE | UC | CCFE | UC | CCFE | UC | CCFE | UC | CCFE | UC |
| Physical functioning | 0.682 | 0.731 | 0.109 | 0.117 | 0.075 | 0.08 | 0.082 | 0.088 | 0.095 | 0.102 | 0.095 | 0.102 |
| Psychological well-being | 0.701 | 0.713 | 0.119 | 0.121 | 0.105 | 0.107 | 0.126 | 0.128 | 0.126 | 0.128 | 0.105 | 0.107 |
| Enjoyment of life | 0.711 | 0.703 | 0.164 | 0.162 | 0.178 | 0.176 | 0.157 | 0.155 | 0.171 | 0.169 | 0.157 | 0.155 |
| Social relationships and participation | 0.72 | 0.694 | 0.058 | 0.056 | 0.065 | 0.062 | 0.079 | 0.076 | 0.072 | 0.069 | 0.072 | 0.069 |
| Resilience | 0.706 | 0.709 | 0.106 | 0.106 | 0.099 | 0.099 | 0.092 | 0.092 | 0.078 | 0.078 | 0.099 | 0.099 |
| Person-centredness | 0.744 | 0.668 | 0.06 | 0.053 | 0.06 | 0.053 | 0.06 | 0.053 | 0.045 | 0.04 | 0.06 | 0.053 |
| Continuity of care | 0.719 | 0.695 | 0.072 | 0.069 | 0.086 | 0.083 | 0.079 | 0.076 | 0.058 | 0.056 | 0.072 | 0.069 |
| Costs (societal care perspective) | 0.673 | 0.739 | 0.02 | 0.022 | 0.04 | 0.044 | 0.04 | 0.044 | 0.054 | 0.059 | 0.047 | 0.052 |
| Overall value scores | | | 0.707 | 0.707 | 0.708 | 0.706 | 0.714 | 0.713 | 0.698 | 0.702 | 0.706 | 0.707 |
| 95% CI* | | | 0.696 to 0.717 | 0.697 to 0.717 | 0.697 to 0.718 | 0.695 to 0.716 | 0.697 to 0.717 | 0.695 to 0.716 | 0.695 to 0.717 | 0.697 to 0.719 | 0.695 to 0.717 | 0.696 to 0.718 |
| % CCFE >UC* | | | 50 | | 57 | | 54 | | 39 | | 48 | |

Colour scheme ranges from red (lowest score) to green (highest score).
*Based on Monte-Carlo simulation.
CCFE, Care Chain Frail Elderly; DCE, discrete choice experiment; UC, usual care.

namely whether or not to include costs in the overall value score.[34 35] To elicit a weight for costs, stakeholders had to trade-off costs against other outcomes, which makes the relative contribution of costs to the overall value score explicit. However, it can be argued that this does not adequately address the opportunity costs of the CCFE.[36] We also performed the MCDA without costs as a sensitivity analysis (see online supplemental appendix 6), which led to higher overall value scores for the CCFE at both time points. However, now the overall value forms a composite benefit score for which a new cost-effectiveness threshold must be determined. Although we believe the current set of outcomes captures the full potential value of an integrated care programme for frail elderly, this set should be tailored to each intervention's aims and target group. Therefore, when the set of outcome measures changes, new thresholds need to be determined. Another option is to calculate the cost-per-value and prioritise interventions with the lowest cost-per-value ratio, but this only leads to a ranking of interventions.[35] A last point of discussion on MCDA is that it requires a deliberative component to avoid making decisions based solely on the model.[35] In our study, we presented results from five stakeholder perspectives which inevitably calls for further deliberation to determine which perspective should prevail.

## Implications

Although the CCFE does not improve the (physical) health of patients, it is still positively evaluated by all stakeholder groups at 6 months. At 12 months stakeholders were mainly indifferent. This warrants further research into interventions to maintain the effects of such programmes in the long-term. Furthermore, we advocate a wider use of MCDA to evaluate multifaceted, person-centred, integrated care programmes for frail elderly that aim to improve multiple outcomes, including those that go beyond health. MCDA enables a transparent and explicit decision-making process and serves as a tool to help decision-makers reach a decision. Therefore, MCDA-results are a good starting point for deliberation before deciding on reimbursement or broader implementation of new interventions.

## CONCLUSION

After 6 months, the overall value score for the CCFE was higher than for usual care across all stakeholders, but at 12 months, the preference for the CCFE had disappeared. The CCFE led to sustained improvements in enjoyment of life and person-centredness, which is aligned with the programme's aim, but also to a deterioration in physical functioning at 6 months and higher costs. Therefore, the CCFE is only the preferred way of delivering care to frail elderly in case improvements in enjoyment of life and person-centredness are considered more important than costs and physical functioning.

**Acknowledgements** We would like to thank all respondents and their informal caregivers for participating in this research. We would also like to thank the care groups DOH, SGE and PoZoB for their continuous support in facilitating this research.

**Contributors** MH, MK, FL, AT and MR-vM developed the overall methodology for the value-based MCDA approach in the SELFIE project. MH, MK, FL and MR-vM contributed to the study design and data collection. MH, MK, LG, KI, AT and MR-vM had access to the data and contributed to the data analysis. MH and MR-vM drafted this manuscript. All coauthors critically reviewed this manuscript. All authors read and approved the final manuscript. MH accepts full responsibility for the finished work and/or the conduct of the study, had access to the data, and controlled the decision to publish.

**Funding** The SELFIE project has received funding from the European Union's Horizon 2020 research and innovation programme under grant agreement No 634288.

**Disclaimer** The content of this manuscript reflects only the SELFIE group's views and the European Commission is not liable for any use that may be made of the information contained herein.

**Competing interests** On submission, MK was affiliated to OPEN Health. No other disclosures were reported.

**Patient and public involvement** Patients and/or the public were involved in the design, or conduct, or reporting, or dissemination plans of this research. Refer to the Methods section for further details.

**Patient consent for publication** Not applicable.

**Ethics approval** The study was conducted as one of the case studies in Horizon 2020-project SELFIE (Sustainable intEgrated care modeLs for multimorbidity: delivery, FInancing and performancE). The study protocol was reviewed by the medical ethics committee of the Erasmus Medical Centre, the Netherlands. The committee concluded that the rules laid down in the Dutch Medical Research Involving Human Subjects Act do not apply to this research (MEC2017-121). All study participants have provided informed written consent.

**Provenance and peer review** Not commissioned; externally peer reviewed.

**Data availability statement** Data are available on reasonable request. The data that support the findings of this study are available on request from the first author, MH. The data are not publicly available due to privacy restrictions.

**ORCID iDs**
Maaike Hoedemakers http://orcid.org/0000-0002-9222-652X
Apostolos Tsiachristas http://orcid.org/0000-0002-4662-8915
Maureen Rutten-van Molken http://orcid.org/0000-0001-8706-3159

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
