## [Reviewer comments · BMJ Open]

ARTICLE DETAILS

TITLE (PROVISIONAL)	Value-based person-centred integrated care for frail elderly living at home: a quasi-experimental evaluation using Multi-Criteria Decision Analysis
AUTHORS	Hoedemakers, Maaïke; Karimi, Milad; Leijten, Fenna; Goossens, Lucas; Islam, Kamrul; Tsiachristas, Apostolos; Rutten-van Molken, Maureen

VERSION 1 – REVIEW

REVIEWER	Lori Weeks Dalhousie University
REVIEW RETURNED	19-Oct-2021

GENERAL COMMENTS	Figure 1 needs to be reformatted to ensure that all details can be viewed. In the discussion, I suggest that the authors add some text about the context (i.e., location) of the where the study occurred and how that had an impact on the results. In addition, a statement about implementing the intervention in other contexts would be useful.
--

REVIEWER	Paul Stolee University of Waterloo, SPHHS
REVIEW RETURNED	02-Nov-2021

GENERAL COMMENTS	This paper reports a carefully conceived and innovative evaluation of an integrated care program – the Care Chain Frail Elderly program (CCFE) for older persons living with frailty. The CCFE, as described, has some admirable qualities, for example, older persons and caregivers are present at team meetings. Financing is done by bundled payment to stimulate collaboration, with a fixed amount of funding/patient. The bundled payments are based on a contract between each care group and the dominant insurer – I am not sure what this means – who are the care groups? It is noted that the top 1% of the “frailtest” older patients are targeted using a case-finding approach. More detail is needed on what that approach is and how it is applied – for example, is a particular screening tool used? Given the clinical and social complexity of frailty, the researchers appropriately recognize the need for multi-faceted measures of outcome and have also considered the Triple Aim framework – thus including cost considerations. Strengths of the paper include input of
--

	frail older adults into the choice of outcome measures, and their weighting of importance from multiple stakeholder perspectives. It is unfortunate that older persons with cognitive impairment were excluded given the apparent restrictions of measuring patient-reported outcomes. Did the authors consider using proxy reports, such as asking a caregiver to complete the measures on the patients' behalf? Proxy reporting has its limitations but seems preferable to excluding a significant group of patients. The use of MCDA is an innovative and rigorous approach to dealing with the complex measurement challenge, while allowing multiple stakeholder perspectives to be reflected. The relative importance weights were elicited in a previous study, with an impressively large sample (n=5122) of stakeholders from eight countries. This study was reported in Rutten-van Mölken et al., 2020 – of particular interest from that report is that patients gave higher weights to physical functioning, and lower weights to person-centredness, than did professionals. This calls into question whether the aim of the CCFE – focused on person-centredness – is well targeted, even though this was an area of sustained benefit in the study. I would like the authors to give some greater consideration to this question, and also to whether the weightings obtained in an abstract discrete choice experiment adequately translate to a real-world application. I suspect that if I had a choice of improved physical function over a person-centred approach, I would choose the former. Over a longer-term period of engagement with a clinical program, I might give a higher weighting to the way that the program respected my values, choices and preferences. Given the program's emphasis on person-centredness, I believe the authors' could have considered using a more explicitly person-centred measure, such as Goal Attainment Scaling, which measures outcomes specifically relevant to individual patients (e.g., Clarkson & Barnett, 2021; Goal attainment scaling to facilitate person-centred, medicines-related consultations - PubMed (nih.gov)), and which has also been used in persons with cognitive impairment (e.g., Jogie et al., 2021; Goal setting for people with mild cognitive impairment or dementia in rehabilitation: A scoping review - PubMed (nih.gov)). Goal Attainment Scaling requires an investment of time, but may have been more feasible than a rather unwieldy battery of standardized measures. Stakeholder engagement throughout the research process is impressive. I have a number of more specific comments which I hope the authors will consider:  - Although I have certainly used this term myself in the past, the term "elderly" has fallen somewhat out of favour. For example, see the position of the American Geriatrics Society https://agsjournals.onlinelibrary.wiley.com/doi/10.1111/jgs.14941. I would leave this decision to the journal's editorial board, but I would recommend using "older persons" rather than "the elderly". And I would recommend that the word "elderlies" never be used, anywhere. - On page eight, frailty is defined "as a loss of functional abilities and control over one's life and in need of complex care". Aside from the awkward syntax, I do not recognize this as a definition with which I
--	---

	am familiar nor as one with which I agree. The authors might wish to consider the definition given by Clegg and colleagues in their Lancet paper Frailty in elderly people - The Lancet or the WHO definition cited in this paper by Muscedere and colleagues Moving Towards Common Data Elements and Core Outcome Measures in Frailty Research SpringerLink  - On page 11, it is stated that costs were obtained with iMTA-Medical Consumption Questionnaire – reference 26 is given for this questionnaire, however that reference makes no mention of a Medical Consumption Questionnaire. - The outcome measures were indicated as having been selected by stakeholders. I was not familiar with a number of these, so am curious how they were identified – were stakeholders choosing from options? - From Table 2 – “After IPW, the matching statistics were within the desired range”, but I did not see a prior statement of what range was targeted. Some more information on interpretation of this table would be helpful, as some of the statistics seem more dissimilar after IPW than before. Thank you for the opportunity to review this interesting manuscript.
--	--

VERSION 1 – AUTHOR RESPONSE

Reviewer: 1

Dr. Lori Weeks, Dalhousie University

Comments to the Author:

1. Figure 1 needs to be reformatted to ensure that all details can be viewed.

Reply: Thank you for noticing. We have enhanced the figure to make sure all text is readable.

2. In the discussion, I suggest that the authors add some text about the context (i.e., location) of the where the study occurred and how that had an impact on the results. In addition, a statement about implementing the intervention in other contexts would be useful.

Reply: Thank you for this suggestion. We have added text about the context of the implementation and how the implementation process may differ in other contexts.

*New text (Discussion, Context and comparison with previous evaluations, line 306-314):
Regarding the context of the CCFE, we should stress that the programme was implemented in a setting with a strong primary care sector. GPs already have a history of collaborating with other primary care providers, for example by working jointly in health centres. Hence, setting up community networks and collaborating in wider multi-disciplinary teams was not such a big step. When implementing similar interventions in countries with a less strong primary care system, collaboration may require more efforts. On the other hand, the potential for savings due to the intervention might be higher in countries where the GPs does not act as a gatekeeper to secondary care, as a programme like the CCFE could substitute more secondary care services by primary care services.*

Reviewer: 2

Dr. Paul Stolee, University of Waterloo

Comments to the Author:

This paper reports a carefully conceived and innovative evaluation of an integrated care program – the Care Chain Frail Elderly program (CCFE) for older persons living with frailty.

1. The CCFE, as described, has some admirable qualities, for example, older persons and caregivers are present at team meetings. Financing is done by bundled payment to stimulate collaboration, with a fixed amount of funding/patient. The bundled payments are based on a

contract between each care group and the dominant insurer – I am not sure what this means – who are the care groups?

Reply: Thank you for the comments on our paper. We understand that the explanation given about care groups was quite limited and have therefore expanded the explanation in the Methods-section.

New text (Methods, Intervention, line 140-142):

These care groups are legal entities of primary care providers who develop chronic care programs, support the provision of these programmes and are contracted by the health insurer to coordinate chronic care in a region.

2. It is noted that the top 1% of the “frailtest” older patients are targeted using a case-finding approach. More detail is needed on what that approach is and how it is applied – for example, is a particular screening tool used?

Reply: The providers of the CCFE have chosen not to apply a particular screening tool. They are convinced that primary care providers, because they know their patients, are better suited to select the elderly that might benefit from the care programme than a screening tool. The health insurer had agreed with this approach. We have added more text about the case-finding approach explaining the top 1% frail elderly.

Additionally, in the description of the intervention it is explained that the nurse practitioner performs a comprehensive geriatric assessment, i.e., a Dutch assessment instrument called ‘TraZAG’. However, this is done to make an inventory of the needs and capabilities, and it is not used as a screening tool.

New text (Methods, Intervention, line 118-122):

A primary care team consisting of the GP, nurse-practitioner, and district nurse, identifies potential candidates for the CCFE using a case-finding approach. To be included in the CCFE, the health insurer does not require a specific diagnosis or the use of a screening tool. They trust that the GP and nurse practitioner know their patients and are most suited to select the elderly that may benefit from the care programme. Furthermore, there is an agreement with the health insurer that only the top 1% frail elderly of a GP-practice will be included in the programme. Subsequently, the nurse-practitioner visits the older person at home and performs a comprehensive geriatric assessment of the needs, capabilities, and preferences in the physical, psychological, cognitive, and social domains.

3. Given the clinical and social complexity of frailty, the researchers appropriately recognize the need for multi-faceted measures of outcome and have also considered the Triple Aim framework – thus including cost considerations. Strengths of the paper include input of frail older adults into the choice of outcome measures, and their weighting of importance from multiple stakeholder perspectives.

Reply: Thanks for recognizing the unique features of this study.

4. It is unfortunate that older persons with cognitive impairment were excluded given the apparent restrictions of measuring patient-reported outcomes. Did the authors consider using proxy reports, such as asking a caregiver to complete the measures on the patients’ behalf? Proxy reporting has its limitations but seems preferable to excluding a significant group of patients.

Reply: We agree that excluding a group of patients is never preferable, however, we believe that including this group using proxy reports would come at the expense of truly representing the older persons’ viewpoints well. Although we considered inviting (informal) caregiver(s) to fill in the questionnaire on the patients’ behalf, we believe that this information would lead to

difficulties and 'noise' in interpreting the results. Also, we wanted to put the perspective of the older persons central. The experience of the informal caregiver(s) might differ from the experience of the patient. A good example of where the experience of a patient and informal caregiver might differ concerns the outcome measure social relationships & participation. An informal caregiver might answer on the patient's behalf that (s)he is not lonely because they visit weekly, whereas the patient may still experience loneliness and prefers to have more social relationships. We did allow informal caregiver(s) to be present during the interview if that was what the older person preferred, but we always tried to avoid that their presence would affect the responses of the older persons.

5. The use of MCDA is an innovative and rigorous approach to dealing with the complex measurement challenge, while allowing multiple stakeholder perspectives to be reflected. The relative importance weights were elicited in a previous study, with an impressively large sample (n=5122) of stakeholders from eight countries. This study was reported in Rutten-van Mölken et al., 2020 – of particular interest from that report is that patients gave higher weights to physical functioning, and lower weights to person-centredness, than did professionals. This calls into question whether the aim of the CCFE – focused on person-centredness – is well targeted, even though this was an area of sustained benefit in the study. I would like the authors to give some greater consideration to this question, and also to whether the weightings obtained in an abstract discrete choice experiment adequately translate to a real-world application. I suspect that if I had a choice of improved physical function over a person-centred approach, I would choose the former. Over a longer-term period of engagement with a clinical program, I might give a higher weighting to the way that the program respected my values, choices and preferences.

Reply: Thanks for this suggestion. We agree that there is a discrepancy between what is important to a patient, and what is feasible in practice. This explains that although a patient would prefer to improve physical functioning, care programmes focus on outcomes such as person-centredness. We have added a reflection upon this in the Discussion section of the manuscript.

New text (Discussion, Context and comparison with other research, line 326-334): One could question the relevance of finding a sustained effect on an outcome (i.e., person-centredness) that was less highly valued than other outcomes in the DCE. This may raise the question whether the aim of the CCFE was well-targeted. However, weights were derived from a DCE that asks respondents to choose between two hypothetical care programmes, which gives them the opportunity to trade-off person-centredness for, for example, improved physical functioning. Even though this is likely to be an appropriate reflection of their preferences if all options were open, in real life this trade-off may no longer exist, because improving physical functioning might not be possible anymore. Hence, there is a discrepancy between what is important to a patient and what is feasible in practice.

6. Given the program's emphasis on person-centredness, I believe the authors' could have considered using a more explicitly person-centred measure, such as Goal Attainment Scaling, which measures outcomes specifically relevant to individual patients (e.g., Clarkson & Barnett, 2021; Goal attainment scaling to facilitate person-centred, medicines-related consultations - PubMed (nih.gov)), and which has also been used in persons with cognitive impairment (e.g., Jogie et al., 2021; Goal setting for people with mild cognitive impairment or dementia in rehabilitation: A scoping review - PubMed (nih.gov)). Goal Attainment Scaling requires an investment of time, but may have been more feasible than a rather unwieldy battery of standardized measures.

Reply: We do agree with the reviewer that Goal Attainment Scaling is a promising approach in measuring person-centeredness. In the current study it was not feasible to include Goal Attainment Scaling, as interviewers did not know the respondents, and only administered the questionnaire and it would be too time consuming. Also, it is worth mentioning that the professionals started piloting Goal Attainment Scaling as part of the care program. To reflect on this a bit more, the instruments used in this study are part of the so-called core set of outcomes of the European SELFIE project, for which we also elicited relative importance weights. In the weight elicitation procedure, it is crucial to very clearly define the outcome measures to ensure that respondents can make trade-offs between the outcome

measures. Goals set in Goal Attainment Scaling can differ largely between patients, and it is therefore not possible to know what the elicited weight represents. Furthermore, it could lead to violating the assumption of preference independence, when the goal of the patient covers one of the other outcomes already included in the core set.

7. Stakeholder engagement throughout the research process is impressive.

Reply: Thanks for acknowledging this.

I have a number of more specific comments which I hope the authors will consider:

8. Although I have certainly used this term myself in the past, the term “elderly” has fallen somewhat out of favour. For example, see the position of the American Geriatrics Society <https://eur03.safelinks.protection.outlook.com/?url=https%3A%2F%2Fagsjournals.onlinelibrary.wiley.com%2Fdoi%2F10.1111%2Fjgs.14941&data=04%7C01%7Cchoedemakers%40eshpm.eu.r.nl%7C922a605bc6f9469fd3208d9a2b7a843%7C715902d6f63e4b8d929b4bb170bad492%7C0%7C0%7C637719731953415483%7CUnknown%7CTWFpbGZsb3d8eyJWljojMC4wLjAwMDAiLCJQIjoiV2luMzliLCJBTiI6Ikk1haWwiLCJXVCi6Mn0%3D%7C1000&data=byctzC8yaVTzHcOFVvWwXcZGutWhsGelEq7LqoTEjCs%3D&reserved=0>. I would leave this decision to the journal’s editorial board, but I would recommend using “older persons” rather than “the elderly”. And I would recommend that the word “elderlies” never be used, anywhere.

Reply: Thanks for this suggestion. We were not aware of the negative connotation of these terms, but this may be a cultural difference between the US and Europe. We prefer to keep the term ‘elderly’ as it is also part of the title of the programme “Care Chain Frail Elderly”, but have changed the term ‘elderlies’ in older persons.

9. On page eight, frailty is defined “as a loss of functional abilities and control over one’s life in need of complex care”. Aside from the awkward syntax, I do not recognize this as a definition with which I am familiar nor as one with which I agree. The authors might wish to consider the definition given by Clegg and colleagues in their Lancet paper Frailty in elderly people - The Lancet or the WHO definition cited in this paper by Muscedere and colleagues Moving Towards Common Data Elements and Core Outcome Measures in Frailty Research | SpringerLink

Reply: We acknowledge that this is an uncommon definition and that there are better scientific definitions. However, it is exactly how the care groups define frailty in the CCFE and represents how the professionals select patients to participate in the care programme. Nevertheless, we have rewritten the sentence slightly to remove the awkward syntax.

New text (Methods, Study population, line 110-111)

The study population consisted of community-dwelling frail elderly, where frailty is defined as being in need of complex care due to loss of functional abilities and control over one’s life.

10. On page 11, it is stated that costs were obtained with iMTA-Medical Consumption Questionnaire – reference 26 is given for this questionnaire, however that reference makes no mention of a Medical Consumption Questionnaire.

Reply: Thanks for noticing. We have replaced the reference with the correct reference: Bouwmans C, Hakkaart-van Roijen L, Koopmanschap M, Krol M, Severens H, Brouwer W. Handleiding iMTA Medical Cost Questionnaire (iMCQ). Rotterdam: iMTA: Erasmus Universiteit; 2013. <https://www.imta.nl/questionnaires/>

11. The outcome measures were indicated as having been selected by stakeholders. I was not familiar with a number of these, so am curious how they were identified – were stakeholders choosing from options?

Reply: The core set of outcomes included in the MCDA was indeed based on focus groups with stakeholders. The focus groups consisted of two parts: (1) defining (A) good health and well-being and (B) a good care process, and (2) discussion on most important concepts and

creating 'top 10' lists. During the first part, we asked participants to complete the sentence 'For me, being in great health means...' and 'I'd be really satisfied with all of the care/the overall care that I receive, if...'. After this, the research group went through the (top 10) statements and tried to move from specific examples to general outcomes. Thus, input for outcome measures came from stakeholders but were sometimes reframed in more conceptual outcomes. Subsequently, researchers searched for instruments that best matched the definitions of the conceptual outcomes. We also wrote a paper about the focus groups: <https://bmjopen.bmj.com/content/8/8/e021072>.

12. From Table 2 – “After IPW, the matching statistics were within the desired range”, but I did not see a prior statement of what range was targeted. Some more information on interpretation of this table would be helpful, as some of the statistics seem more dissimilar after IPW than before.

Reply: We agree with the reviewer that this may not be clear to the reader. What we meant with 'within desired range' is line 188-189 "For sufficient balance, B should be less than 25% and R be between 0.5 and 2." To clarify, we have added this information to Table 2. Furthermore, although some statistics are more dissimilar after IPW, overall the matching statistics have improved.

*New text (Results, Respondents, line 243-244):
After IPW, the matching statistics were within the desired range (Rubin's B < 25%, Rubin's R 0.5 – 2).*

VERSION 2 – REVIEW

REVIEWER	Paul Stolee University of Waterloo, SPHHS
REVIEW RETURNED	02-Dec-2021

GENERAL COMMENTS	I appreciate the thoughtful responses from the authors, and believe the revisions have adequately addressed the points raised in the previous review. My only comment is re point 2. It is helpful to have the extra detail on the rationale for the case-finding approach used to identify the top 1%. While I understand the rationale for the approach, I think it would be appropriate to acknowledge potential limitations of this approach, including that the generalizability of the approach may be limited by the ability of clinicians at other sites to identify a similar group of at-risk patients.
---

VERSION 2 – AUTHOR RESPONSE

Reviewer: 2

Dr. Paul Stolee, University of Waterloo

Comments to the Author:

I appreciate the thoughtful responses from the authors, and believe the revisions have adequately addressed the points raised in the previous review. My only comment is re point 2. It is helpful to have the extra detail on the rationale for the case-finding approach used to identify the top 1%. While I understand the rationale for the approach, I think it would be appropriate to acknowledge potential limitations of this approach, including that the generalizability of the approach may be limited by the ability of clinicians at other sites to identify a similar group of at-risk patients.

Reply: We would like to thank the reviewer for his thorough review. As per his comment, we have added the potential limitations of the use of a case-finding approach to the Discussion section of the manuscript.

New text (Discussion, Strengths & limitations, line 337-xx):

In the current study, a potential limitation concerns the case finding approach to identify the target population of both groups, namely frail elderly in need of complex care due to loss of functional abilities and control over one's life. In the intervention group this was done by GPs offering the programme and in the control group by GPs not offering the programme. To ensure a similar level of frailty in both groups, the latter GPs were assisted by a GP specialised in elderly care. This has been successful as the baseline characteristics of both groups were quite similar.